# Bismuth Doping in Nanostructured Tetrahedrite: Scalable Synthesis and Thermoelectric Performance

**DOI:** 10.3390/nano11061386

**Published:** 2021-05-25

**Authors:** Peter Baláž, Emmanuel Guilmeau, Marcela Achimovičová, Matej Baláž, Nina Daneu, Oleksandr Dobrozhan, Mária Kaňuchová

**Affiliations:** 1Institute of Geotechnics, Slovak Academy of Sciences, 04001 Košice, Slovakia; balaz@saske.sk (P.B.); balazm@saske.sk (M.B.); 2CRISMAT, CNRS, Normandy University, ENSICAEN, UNICAEN, 14000 Caen, France; emmanuel.guilmeau@ensicaen.fr; 3Jozef Stefan Institute, SI-1000 Ljubljana, Slovenia; nina.daneu@ijs.si; 4Department of Electronics and Computer Technology, Sumy State University, 40007 Sumy, Ukraine; dobrozhan.a@gmail.com; 5Institute of Earth Resources, Technical University Košice, 04001 Košice, Slovakia; maria.kanuchova@tuke.sk

**Keywords:** tetrahedrite, doping, bismuth, high-energy milling, thermoelectricity

## Abstract

In this study, we demonstrate the feasibility of Bi-doped tetrahedrite Cu_12_Sb_4−x_Bi_x_S_13_ (x = 0.02–0.20) synthesis in an industrial eccentric vibratory mill using Cu, Sb, Bi and S elemental precursors. High-energy milling was followed by spark plasma sintering. In all the samples, the prevailing content of tetrahedrite Cu_12_Sb_4_S_13_ (71–87%) and famatinite Cu_3_SbS_4_ (13–21%), together with small amounts of skinnerite Cu_3_SbS_3_, have been detected. The occurrence of the individual Cu-Sb-S phases and oxidation states of bismuth identified as Bi^0^ and Bi^3+^ are correlated. The most prominent effect of the simultaneous milling and doping on the thermoelectric properties is a decrease in the total thermal conductivity (*κ*) with increasing Bi content, in relation with the increasing amount of famatinite and skinnerite contents. The lowest value of *κ* was achieved for x = 0.2 (1.1 W m^−1^ K^−1^ at 675 K). However, this sample also manifests the lowest electrical conductivity *σ*, combined with relatively unchanged values for the Seebeck coefficient (*S*) compared with the un-doped sample. Overall, the lowered electrical performances outweigh the benefits from the decrease in thermal conductivity and the resulting figure-of-merit values illustrate a degradation effect of Bi doping on the thermoelectric properties of tetrahedrite in these synthesis conditions.

## 1. Introduction

In the Cu-Sb-S system, several ternary copper sulphides like chalcostibnite CuSbS_2_, skinnerite Cu_3_SbS_3_, famatinite Cu_3_SbS_4_ and tetrahedrite Cu_12_Sb_4_S_13_ exist [1]. Among them, tetrahedrite, an Earth-abundant copper sulphide mineral with extraordinary thermoelectric properties [2,3], shows a continuous interest in the thermoelectric community [4,5,6,7,8,9,10].

The enhancement of thermoelectric performances can often be accomplished via defect engineering. In this approach, two strategies to modify/generate defects in solids are applied, i.e., band engineering and phonon engineering [11]. In the former, the power factor *S*^2^*σ* (*S* is the Seebeck coefficient, and *σ* the electrical conductivity) is enhanced by electronic band structure engineering, while in the second case, the lattice thermal conductivity *κ_L_* is reduced by enhanced phonon scattering [11,12,13]. Transport properties *S*, *σ* and *κ* are embodied in the dimensionless figure-of-merit *ZT* = *S*^2^*σ*/*κ* (where *T* is temperature and *κ* is a sum of lattice thermal conductivity *κ_L_* and electronic thermal conductivity *κ_e_*). Based on the macroscopically measurable parameters *S*, *σ*, and *κ*, the calculated figure-of-merit illustrates the efficiency of a thermoelectric material [13]. To ensure high voltage output and low Joule losses, high *S* and *σ* are needed. To maintain the temperature gradient between the hot and cold sides of a thermocouple, a low *κ* value is required [14]. Thermoelectric devices are highly desirable because of their silent operation, reliability, scalability, predictability and durability. They are particularly appreciated when cost and energy efficiency are not as important as energy availability [15]. However, significantly lower manufacturing costs, more use cases and increased performances are required to meaningfully convert waste heat and reduce fossil fuels consumption [9].

Since the 1950s, doping of prospective thermoelectric materials has been the most popular way to enhance *ZT* values [14]. Generally, doping affects the charge carrier concentration as well as serving as point-defects to enhance phonon scattering. In the 1990s, nanostructure engineering was successfully applied and later used as an alternative to enhance *ZT* values [11,13,15,16,17,18,19,20]. Moreover, when non-equilibrium procedures, such as high-energy milling (HEM), melt spinning (MS) and self-sustaining heating synthesis (SHS), is applied, the delicate multiscale structures/defects can be obtained [13]. In the case of HEM [21,22,23,24,25,26,27,28,29] combined with spark plasma sintering (SPS), the multiscale dimensionality reduction and the formed nanostructures also contribute to phonon scattering and thus, to a decrease in the lattice thermal conductivity [20].

Doping of tetrahedrite Cu_12_Sb_4_S_13_ has been performed in numerous studies, see e.g., references in [6,10]. While Cu and S positions have been doped many times, doping on Sb site has been slightly overlooked. In mineral Cu_12_Sb_4_S_13_ this site is usually partly occupied by As and/or Bi atoms [30,31]. While As is a toxic element, Bi has recently expanded toward engineering fields and nanomedicine thanks to its efficiency and non-toxicity [32]. In synthetic Cu_12_Sb_4_S_13_, several attempts have been made to optimise its thermoelectric performance via doping with Bi. Several synthesis strategies to modify Cu_12_Sb_4_S_13_ thermoelectric properties with Bi doping were applied [33,34,35,36]. For instance, melt-spinning method for samples preparation by application of the Taguchi method has been successfully applied, which is a statistical technique to make robust design of experiments. Moreover, the oxidation and corrosion studies revealed the possibility to use doped tetrahedrite materials under extreme conditions [35]. The simultaneous double- or triple-doping of tetrahedrite to obtain Cu_12−x_Ni_x_Sb_4−y_Bi_y_S_13−z_Se_z_ compounds was also studied. However, this approach complicated the elucidation of the Bi effect itself. The single- and double-doping was also performed in [37,38] where Zn and Bi dopants were used. For Cu_12−x_Zn_x_Sb_4−y_Bi_y_S_13_ synthesis, HEM and HP methods were applied in this case. As mentioned in the work of Goncalves et al., other phases like famatinite and skinnerite were detected. Kumar et al. synthesised Bi doped tetrahedrites with nominal composition of Cu_12_Sb_4−x_Bi_x_S_13_ (x = 0, 0.2, 0.4, 0.6, 0.8) by high temperature solid state reaction method [39,40]. The best results of thermoelectric performance were achieved for samples with the lowest content of bismuth.

Based on the literature, we have found several issues which were not fully investigated. First, as stated in [39], increasing Bi content in the interval (0.2–0.8) leads to the deterioration of thermoelectric properties of tetrahedrite. The highest power factor and figure of merit was obtained for the composition Cu_12_Sb_3.8_Bi_0.2_S_13_ where the lowest content of Bi was applied. The region with Bi content lower than 0.2 [39] and 0.1 [38] was not examined at all. Secondly, there is no evidence in the literature of the oxidative states and/or in which compounds the Bi dopant is present in the Cu-Sb-Bi-S system. Finally, the simultaneous synthesis and doping in an industrial mill which could document sca-ling possibility of thermoelectric materials preparation was not performed for this system up until now.

## 2. Materials and Methods

For mechanochemical synthesis of doped tetrahedrite Cu_12_Sb_4−x_Bi_x_S_13_(x = 0.025, 0.05, 0.1, 0.15, 0.2) the following precursors were used: copper (Merck, Darmstadt, Germany, 99% purity, 99% particles below 70 μm), antimony (Alfa Aesar, Kandel, Germany, 99.5% purity, 99% particles below 120 μm), bismuth (Sigma Aldrich, Taufkirchen, Germany, 99.99% purity, 99% particles below 152 μm) and sulphur (CG-Chemikalien, Laatzen, Germany, 99% purity, 99% particles below 390 μm).

Mechanochemical solid-state syntheses were carried out in an industrial eccentric vibratory ball mill ESM 656–0.5ks (Siebtechnik, Mülheim an der Ruhr, Germany) working under the following conditions: 5 L steel satellite milling chamber attached to the main corpus of the mill, tungsten carbide balls with a diameter of 35 mm, and total mass of 30 kg, 80% ball filling in milling chamber, amplitude of the inhomogeneous vibration 20 mm, rotation speed of the eccenter 960 min^−1^, argon atmosphere. The total feed of reaction precursors was 100 g per batch. Bi was added in the amounts 0.025, 0.05, 0.1, 015 and 0.2 atoms per 4 atoms of Sb in tetrahedrite formula (X_Bi_). The milling was performed for 1 h for all samples. The photograph of the mill is shown in ESI (Appendix A). After completion of the milling programs, the resulting pulverised powders were shaped and densified using SPS (FCTHPD25, Rauenstein, Germany) at 723 K for 30 min (heating and cooling rate of 100 K min^−1^) under a pressure of 64 MPa using graphite dies of 10 mm diameter and a slight over pressure of 30 hPa (Ar), in order to prevent sulphur volatilisation. The final thickness of the pellets was around 8 mm with a geometrical density above of 95% of the crystallographic value.

The qualitative identification of the phase composition of the sintered samples was performed by XRD method with an X´Pert PW 3040 MPD diffractometer (Phillips) wor-king in the 2θ geometry with CuK_α_ radiation. XRD patterns of as-received (milled) samples were collected using a D8 Advance diffractometer (Bruker, Karlsruhe, Germany) with the CuK_α_ radiation in the Bragg–Brentano configuration. The generator was set up at 40 kV and 40 mA. The divergence and receiving slits were 0.3° and 0.1 mm, respectively. The XRD patterns were recorded in the range of 2θ = 10–80° with a step of 0.05°. Rietveld refinements of XRD data of the as-prepared samples were performed using Diffrac^plus^ TOPAS software (version 6, Bruker, Karlsruhe, Germany). The JCPDS-PDF database was used for phase identification [41].

XPS spectrometer SPECS PHOIBOS 100 SCI (SPECS Surface Nano Analysis GmbH Berlin, Germany) and non-monochromatic X-ray source were used. The core spectra were measured at 70 eV and the survey spectrum at 30 eV at room temperature. Basic pressure was 1.10^−8^ mbar. Al K_α_ excitation at 10 kV for all spectra were acquired. For data analysis, SPECSLab2 CasaXPS software (CasaSoftware Ltd., Teignmouth, UK) was used. All peaks were fitted with Shirley and Tougaard type baseline. Silver (Ag3d) was used for calibration of the spectrometer. Charging of samples was resolved by the calibration on carbon.

The sample with the highest Bi addition (X_Bi_ = 0.2) was investigated using scanning and transmission electron microscopy (SEM and TEM). SEM analyses of polished cross-section were performed on a field-emission source microscope (FEG-SEM; JSM-7600F, Jeol Ltd., Tokyo, Japan) operated equipped with an energy dispersive X-ray spectrometer (EDXS; INCA Oxford 350 EDXS SDD, Oxfordshire, UK). Samples for SEM/EDXS analyses were prepared by grinding and polishing, finally using 3-micron diamond lapping film. TEM/EDXS analyses were performed on a conventional microscope operated at 200 kV (JEM 2100, Jeol Ltd., Tokyo, Japan) equipped with EDXS spectrometer. For TEM analyses, smaller fragments of the sintered compacts were initially fixed between two silicon supports using epoxy resin. Then, the composite was mounted into brass tube with 3-mm diameter and prepared further using the conventional approach including thinning, dimpling and finally ion-milling (PIPS 691, Gatan Inc., Pleasanton, CA, USA) using 3.8 kV Ar+ ions at an incidence angle of 10° until perforation. Prior to TEM analyses, the specimens were coated by 3 nm layer of carbon to improve surface electron conductivity (PECS 68s, Gatan, Pleasanton, CA, USA). The electrical resistivity ρ and Seebeck coefficient S were measured simultaneously from ingots, from 300 K up to 700 K using an ZEM-3 (Ulvac-Riko, Ikonobe-cho, Japan) under partial helium pressure. LFA 457 apparatus (Netzsch, Selb, Germany) was used to measure the thermal diffusivity under argon flow. The thermal conductivity *κ* was determined as the product of the geometrical density, the thermal diffusivity, and the theoretical heat capacity (Dulong−Petit approximation). The Wiedemann-Franz law, using a Lorenz number estimated from the relationship L = 1.5 + exp(−|S|/116) [42] was used to calculate the lattice thermal conductivity by subtracting the electronic contribution from the total thermal conductivity (*κ_L_* = *κ* − *κ_e_*). The estimated measurement uncertainties are 6% for the Seebeck coefficient, 8% for the electrical resistivity, 11% for the thermal conductivity, and 16% for the final figure of merit [43].

## 3. Results

### 3.1. Phase Analysis and Structural Parameters

XRD patterns of the sintered samples are displayed in Figure 1. The main phase corresponds to tetrahedrite Cu_12_Sb_4_S_13_ (space group of *I*$\bar{4}$3m), with a large proportion of famatinite Cu_3_SbS_4_ clearly visible. These two phases are present in all Bi-doped samples. The broad diffraction peaks indicate that the crystallised domains are rather small, estimated from Rietveld refinement at 72–119 nm and 61–114 nm for tetrahedrite and famatinite, respectively. Traces of skinnerite Cu_3_SbS_3_ were only detected for the most heavily Bi-doped samples (x = 0.15–0.2). In these samples, the presence of Cu_4_Bi_4_S_9_ phase can be also hypothesised in agreement with the results by Chen et al. [44]. This multiphase behaviour can be suppressed in favour of tetrahedrite by cationic doping on Cu position (see e.g., References in 10) and/or by prolonged high energy milling with subsequent SPS treatment [45].

The quantitative phase relations among the synthesised Cu-Sb-S phases determined by the Rietveld refinement are shown in Figure 2. For x = 0.025, a maximum content of 87% tetrahedrite is observed, with 21% of famatinite and 8% of skinnerite. These estimated values are in agreement with the Gibbs free energy values, which measures the feasibility of solid phase formation: i.e., ∆G = −917.2 kJ mol^−1^, −266.6 kJ mol^−1^ and −216.9 kJ mol^−1^ for tetrahedrite, famatinite and skinnerite, respectively [46]. The influence of Bi on Sb substitution on the phase distribution is clearly demonstrated. With increasing the amount of bismuth, more famatinite and skinnerite phases are formed at the expense of tetrahedrite. This is in agreement with the results of triple cationic (including Bi) and anionic substitution of high-temperature synthesised tetrahedrite published by Goncalves et al. [33]. Famatinite and skinnerite phase formation was also previously observed during the synthesis of tetrahedrites [37,39]. In our previous work on pristine tetrahedrite [10], the formation of famatinite at the expense of tetrahedrite was observed only for long milling durations. In analogy with a high temperature treatment [47], this phase can be formed by the reaction of elemental sulphur with tetrahedrite. In the present study, the phase formation relationship is more complex due to the presence of bismuth. It remains questionable whether Bi remains in its elemental form and/or is substituted in tetrahedrite and/or famatinite as Bi^3+^ or Bi^5+^ oxidation state. For example, the reaction of bismuth with sulphur to form Bi_2_S_3_ is thermodynamically feasible (∆G = −139.3 kJ mol^−1^) [48] and therefore cannot be excluded at least as a reaction intermediate. In agreement with Chen et al. [49], the occurrence of Bi^3+^ in Cu_4_Bi_4_S_9_ structure can be hypothesised. The identification of bismuth oxidation states in the reaction products is further elucidated by using the more sensitive method of X-ray Photoelectron Spectroscopy (XPS), see Section 3.2.

The values for the unit cell volume V_T_ and V_F_ of tetrahedrite Cu_12_Sb_4_S_13_ and famatinite Cu_3_SbS_4_, respectively are displayed in Figure 3 as a function of the Bi content X_Bi_. Unambiguously, a monotonous decrease of the parameter V_T_ for tetrahedrite is observed for X_Bi_ = 0.025–0.10 with a subsequent irregular increase for X_Bi_ = 0.15–0.20. The necessity to apply higher Bi content for cell expansion was reported by Kumar et al., who observed an increase in tetrahedrite lattice parameter *a* for X_Bi_ = 0.2–0.8 [39]. The irregular increase of the lattice constant for Bi-doped tetrahedrite was also documented in [38]. The situation for famatinite is much more straight-forward: starting from X_Bi_ = 0.05, the parameter V_F_ continuously increases. To summarize, the lattice expansion is documented for both phases at the higher Bi content (X_Bi_ = 0.15–0.20).

Generally, the lattice expansion is a consequence of an external intervention into the crystal structure, e.g., by high energy milling and/or atomic substitution [50]. Both approaches were applied in our case: milling was performed constantly for one hour, and Bi was incorporated in various amounts. The ionic radii of Bi^3+^ and Bi^5+^ are equal to 96–117 nm (depending on coordination number) and 76 nm, respectively [51]. Antimony site (where bismuth has to be incorporated) is in Sb^3+^ state for tetrahedrite and skinnerite and in Sb^5+^ state for famatinite [6]. The corresponding values of Sb ionic radii are 76 nm for tetrahedrite and skinnerite and 60 nm for famatinite, respectively. To elucidate the pre-sence of oxidation states of Bi and Sb in the synthesised samples, X-ray photoelectron spectroscopy (XPS) analyses were performed. 

### 3.2. XPS Analysis

The XPS patterns were investigated in detail and are displayed in Figure 4. The spectra for Sb 3d and Bi 4f were recorded in a high-resolution core level mode. Based on the binding energy values, a peak splitting (PS) was also determined. The following oxidation states can be expected: Sb^3+^ for tetrahedrite and skinnerite [6], Sb^5+^ for famatinite [52] and Bi^3+^ and Bi^5+^ for Bi-doped samples.

XPS spectra for antimony are depicted in Figure 4a for the sintered sample with no-minal Bi content, X_Bi_ = 0.2. Two intensive peaks corresponding to 3d_5/2_ and 3d_3/2_ states are present. Peak at E_b_ = 539.82 eV can be connected with Sb^3+^ state corresponding to antimony oxidation state in tetrahedrite and skinnerite [10,53]. Peak at 530.82 eV corresponding to Sb^5+^ is related to famatinite [52], which is present in 21% as determined by XRD (see Figure 2).

XPS spectra for bismuth are depicted for the same sample in Figure 4b. The peaks for Bi4f_5/2_ and Bi4f_7/2_ are shown. Peak-splitting value PS = 5.2 eV deduced from E_b_ = 159.38 eV and E_b_ = 164.58 eV respectively is characteristic of Bi^3+^ oxidation state [52,54]. However, bismuth in Bi^3+^ state for these two peaks creates only ~62% of the total peak area. In the centre of XPS Bi4f spectrum, the third peak is also present. This peak corresponds to non-consumed bismuth in zero-valent form (Bi^0^) and accounts for the rest of the total peak area.

### 3.3. Microstructural Analyses with SEM and TEM

Backscattered images of the sample with X_Bi_ = 0.2 recorded at lower and higher magnification (Figure 5a,b) show that the sample is composed of three main phases with different grey shades implicating different average Z-values (Z_skinnerite_ > Z_tetrahedrite_ > Z_famatinite_). Chemical composition analyses with SEM/EDS (Figure 5c) have confirmed that the main phase (matrix) with medium grey contrast is tetrahedrite (Cu_12_Sb_4_S_13_), grains with the darkest contrast are famatinite Cu_3_SbS_4_, whereas the grains with the brightest contrast are skinnerite Cu_3_SbS_3_. The three phases are fairly homogenously distributed in the sample (Figure 5a) and, at first view, it appears that the average size of grains is in the micron-range, i.e., significantly larger than the diffraction domain size determined from Rietveld refinement. A closer observation of the microstructures recorded at higher magnifications (Figure 5b) reveals that some areas with uniform colour that belong to the same phase are typically composed of smaller grains with sizes well below micron as expected with nano-sized pores at the grain boundaries due to the SPS processing. In addition, nano-sized inclusions with brighter contrast are typically observed within the tetrahedrite matrix. SEM/EDS has shown the presence of a small fraction of Bi only in spectra recorded from skinnerite-rich areas, whereas signal from Bi was below SEM/EDS detection limit in tetrahedrite and famatinite areas. The presence of Bi in skinnerite particles is in agreement with the results of XRD and XPS which showed that the presence of Bi enhances the formation of skinnerite where Bi is in Bi^3+^ form. Quantification of Bi in SEM/EDS spectra from skinnerite grains showed around 3–4 at% of Bi, however, these results probably underestimate the amount of Bi in skinnerite due to the small (μm^3^) interaction volume that also includes grains without (or with a significantly lower amount of) Bi in the analysis.

Besides areas with uniform phase distribution (as shown in Figure 5a,c), SEM ana-lyses revealed few remnants of unreacted initial Sb- and Sb/Bi-rich initial particles with typical diffusion-type microstructure around these particles that developed during SPS. The presence of elemental Bi from XPS most probably stems from these unreacted par-ticles (see ESI, Appendix A, Appendix A).

The sample was analysed with TEM, depicted in Figure 6a, showing a typical situation where a skinnerite grain is surrounded by tetrahedrite grains and nano-sized pores trapped at the grain boundaries. Famatinite grains were not found in the TEM sample indicating that the sample was most likely prepared from the tetrahedrite matrix region, where the presence of brighter nanoparticles is observed in SEM. Selected area electron diffraction (SAED) analysis of a tetrahedrite grain oriented along the [111] zone axis is shown in Appendix A (see ESI). TEM/EDS analysis was used to analyse chemical composition of the phases. It has been shown that skinnerite contains in average 43.0 at% of S, 42.8 at% of Cu, 9.5 at% of Sb and 4.7 at% of Bi. The analyses indicate that Bi replaces Sb at its regular sites in skinnerite, as expected. TEM/EDS analyses did not unambiguously confirm the presence of Bi in tetrahedrite, as the amount of Bi was at the detection limit, i.e., below 1 at%.

### 3.4. Thermoelectric Performance

The temperature dependence of the electrical conductivity (*σ*), Seebeck coefficient (*S*) and power factor (*PF* = *S*^2^*σ*) collected over the temperature range of 300–700 K is shown in Figure 7. The electrical conductivity, *σ*, increases with temperature up to 500 K with a subsequent decrease, while the Seebeck coefficient *S* increases over the full temperature range. The positive sign of *S* confirms *p*-type carrier of all Bi-doped samples. The decrease in the electrical conductivity with increasing Bi content can be explained by the rising presence of famatinite [49]. Moreover, the high thermal conductivity of famatinite contributes further to the loss of thermoelectric performance [55]. Overall, as the amount of “non-tetrahedrite phases” increases with Bi content, see Figure 2, the sample with the lowest Bi content (X_Bi_ = 0.025) exhibits the highest values of *σ*, *S* and consequently *PF*. The sample doped with the highest Bi content (X_Bi_ = 0.2) showed the lowest values of electrical conductivity (3.7 × 10^4^ S m^−1^ at 300 K, 5.2 × 10^4^ S m^−1^ at 500 K and 4.4 × 10^4^ S m^−1^ at 700 K). It should be noted that the increase of Bi content does not affect much the values of Seebeck coefficient which are equal to 87 μV/K at 300 K and 142 μV K^−1^ at 700 K. This suggests that bismuth in tetrahedrite remains in 3+ oxidation state. The U shape of the electrical conductivity is classically observed for tetrahedrites presenting the same magnitude of Seebeck coefficient (i.e., carrier concentration) [3,4]. The material is here in an intermediate regime between a degenerate semiconductor and semiconductor.

Therefore, the relatively high values of PF are achieved for the sample with the hi-ghest Bi content (X_Bi_ = 0.2) which reaches 0.28 mW m^−1^ K^−2^ at 300 K and 0.85 mW m^−1^ K^−2^ at 700 K. Such values are in good agreement with other works on Bi-doped tetrahedrites [34,39]. As the carrier concentration seems unchanged with varying Bi content (similar Seebeck coefficient), the decrease of PF values for Bi-substituted samples are most likely due to additional secondary phases (famatinite, skinnerite) acting as scattering barriers for the mobility of carriers.

The temperature dependences of the total (*κ*) and lattice (*κ_L_*) thermal conductivity are shown in Figure 8. The decrease in *κ* as the Bi content increases is mainly explained by the decrease of the electronic contribution caused by a reduced electrical conductivity.

To summarise, the temperature dependence of the figure of merit, *ZT*, is the highest for undoped Cu_12_Sb_4_S_13_ tetrahedrite (*ZT*~0.67 at 700 K) suggesting a degradable effect of Bi doping on the thermoelectric performances of the explored system due to the decrease in electrical conductivity.

## 4. Conclusions

Elemental precursors Cu, Sb, Bi and S were successfully used to synthesise Bi-doped tetrahedrite Cu_12_Sb_4−x_Bi_x_S_13_ (x = 0.02–0.20) in an industrial mill. High-energy milling, in duration of one hour followed by spark plasma sintering, led to several phases like tetrahedrite, famatinite and skinnerite, and their nanoscale dimensions have been determined. Among them, tetrahedrite Cu_12_Sb_4_S_13_ is prevailing with content up to 87% as determined by Rietveld refinement. Special attention was devoted to Bi presence which was doped in amounts x = 0.02–0.20 for Cu_12_Sb_4−x_Bi_x_S_13_. Using XPS and SEM/EDS methods, Bi was found in +3 oxidation state and mainly concentrated in skinnerite phase where it acts as a stabilising agent. However, SEM analyses revealed few remnants of unreacted Sb- and Sb/Bi-rich particles with typical diffusion-type microstructure around these particles that deve-loped during SPS. The presence of elemental Bi detected by XPS most probably stems from these unreacted particles.

The most prominent effect of the simultaneous milling and doping on the thermo-electric properties is a decrease in the total thermal conductivity (*κ*) with increasing Bi content, in relation with the increasing amount of famatinite and skinnerite contents. The lowest value of *κ* was achieved for x = 0.2 (1.1 W m^−1^ K^−1^ at 675 K). However, this sample also manifests the lowest electrical conductivity, *σ*, along with unchanged values for Seebeck coefficient (*S*) resulting in a decrease in the figure-of-merit values. This illustrates the degradation effect of Bi doping on the thermoelectric properties of tetrahedrite, as well as demonstrating that complex microstructures are not always beneficial for the thermoelectric performance. A trade-off between electrical conductivity and Seebeck coefficient must be considered in order to retain a competitive power factor and thus benefit from a lower thermal conductivity. In the present case, significant amounts of second phases are very detrimental to the thermoelectric performances despite interesting microstructural features.

## Figures and Tables

**Figure 1 nanomaterials-11-01386-f001:**
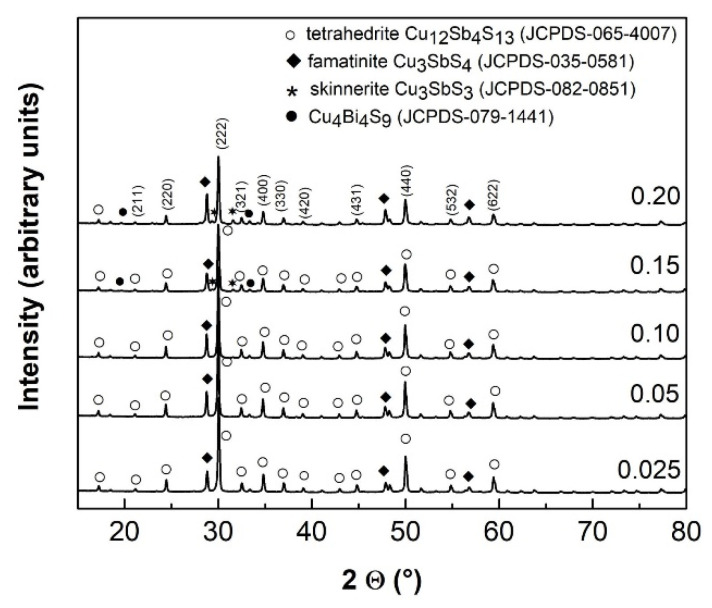
XRD patterns of sintered samples with nominal Bi content, X_Bi_ = 0.025, 0.05, 0.10, 0.15 and 0.20.

**Figure 2 nanomaterials-11-01386-f002:**
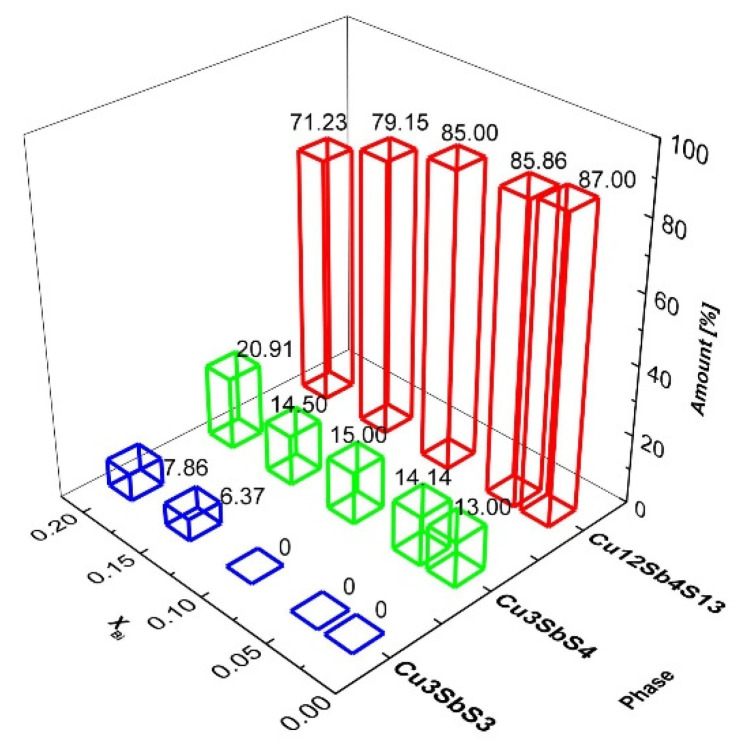
Phase distribution of tetrahedrite Cu_12_Sb_4_S_13_, famatinite Cu_3_SbS_4_ and skinnerite Cu_3_SbS_3_ for sintered samples as a function of the nominal Bi content, X_Bi_.

**Figure 3 nanomaterials-11-01386-f003:**
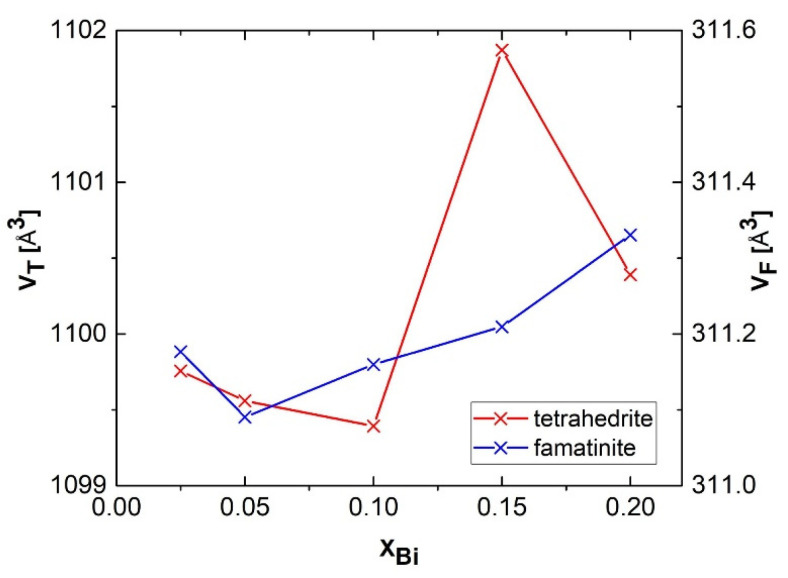
Unit cell volume of tetrahedrite Cu_12_Sb_4_S_13_, V_T_ and famatinite Cu_3_SbS_4_, V_F_ for sintered samples as a function of the nominal Bi content, X_Bi_.

**Figure 4 nanomaterials-11-01386-f004:**
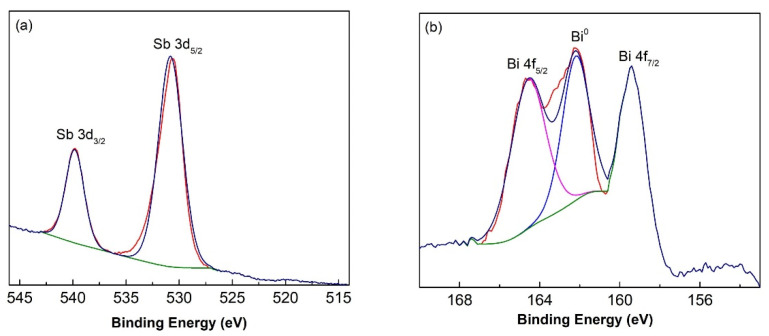
High resolution XPS spectra of antimony Sb 3d (**a**) and bismuth Bi 4f (**b**) for sintered sample with nominal Bi content, X_Bi_ = 0.2.

**Figure 5 nanomaterials-11-01386-f005:**
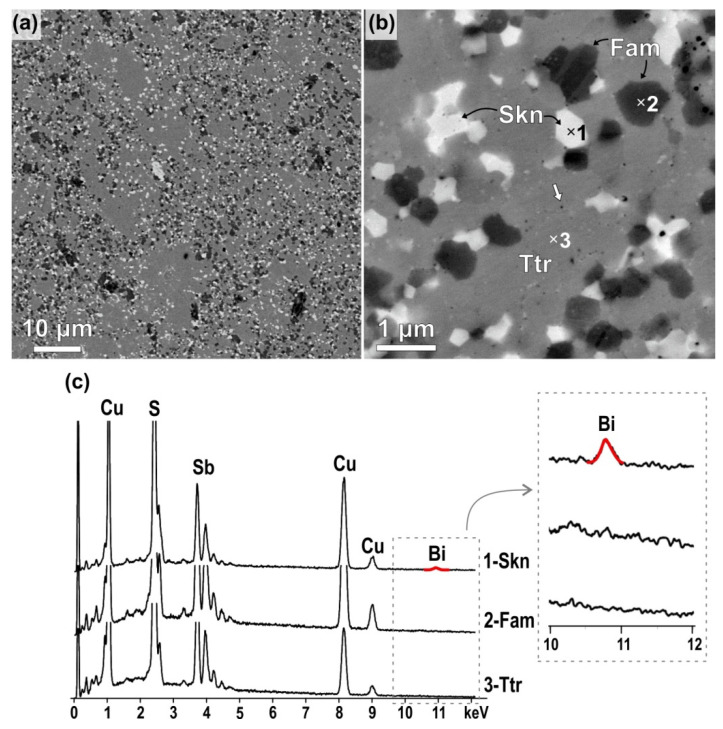
Backscattered SEM images of the sample with a nominal Bi content, X_Bi_ = 0.2 at (**a**) lower and (**b**) higher magnification revealing the presence of three main phases with different average atomic densities (Z). (**c**) EDS spectra recorded from areas with different contrasts reveal detectable amounts of Bi only in skinnerite.

**Figure 6 nanomaterials-11-01386-f006:**
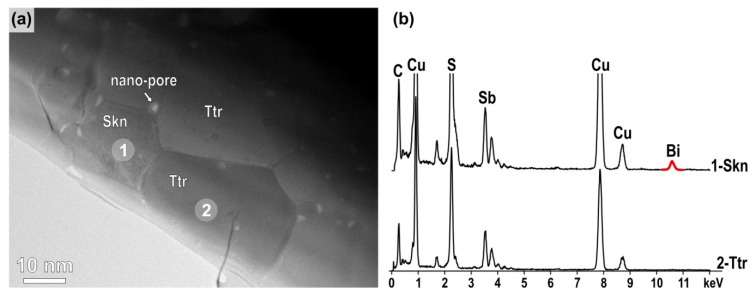
(**a**) TEM image and (**b**) EDS analysis of the X_Bi_ = 0.2 sample show that Bi is preferentially incorporated into skinnerite.

**Figure 7 nanomaterials-11-01386-f007:**
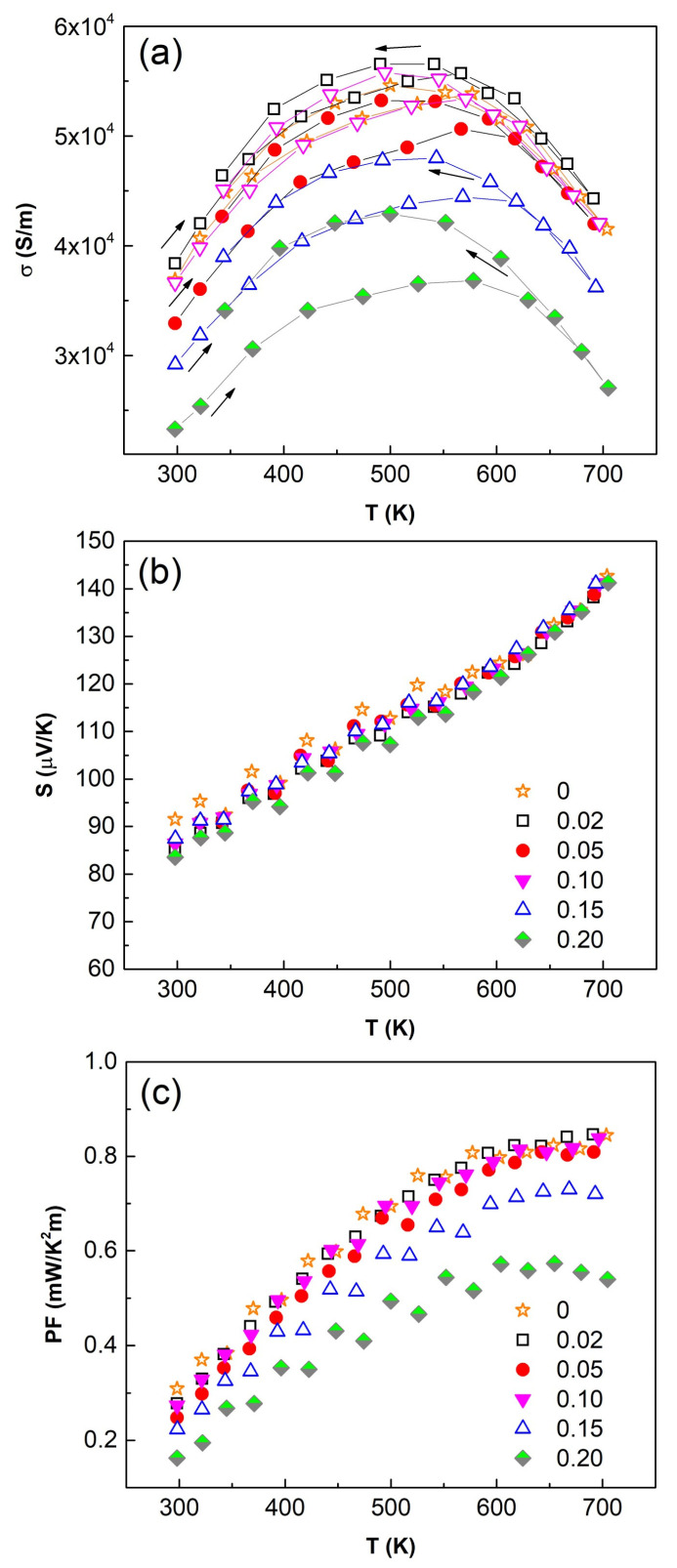
Temperature dependence of (**a**) electrical conductivity *σ*, (**b**) Seebeck coefficient *S*, and (**c**) power factor *PF* of sintered samples. Nominal Bi content, X_Bi_: see Figure.

**Figure 8 nanomaterials-11-01386-f008:**
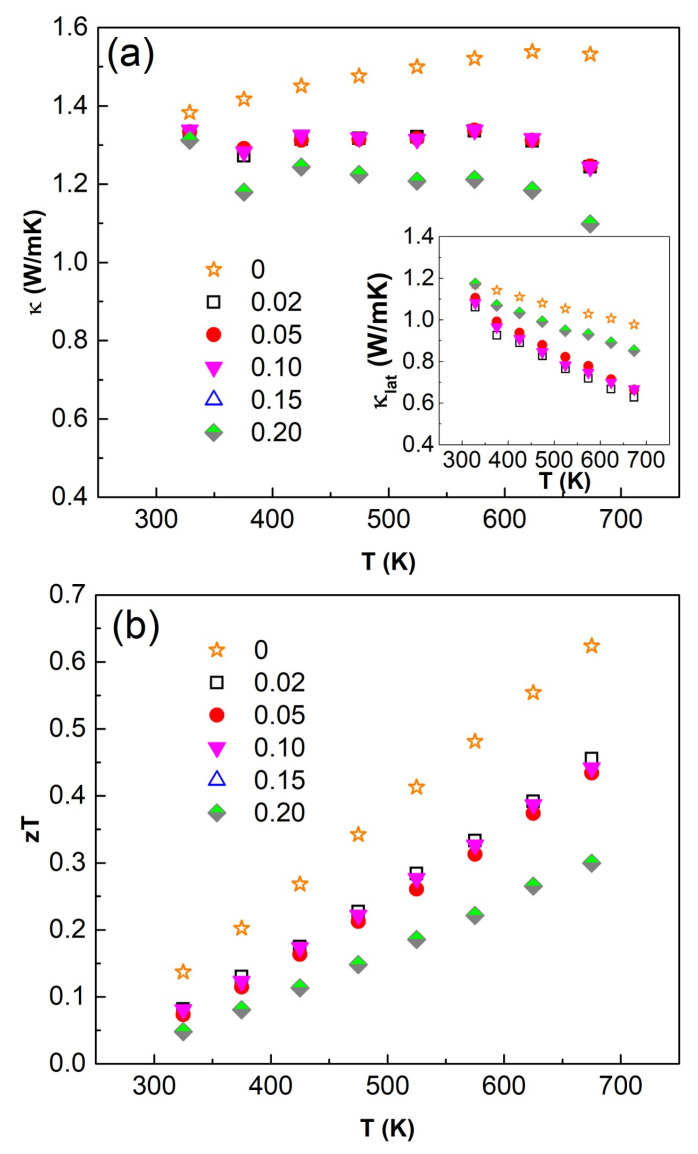
Temperature dependence of (**a**) thermal conductivity, and (**b**) dimensionless figure of merit, *ZT* of sintered samples as a function of nominal Bi content, X_Bi_: see Figure. Inset shows the lattice component, *κ_lat_* of the total thermal conductivity.

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
