# Peer review of "Bismuth Doping in Nanostructured Tetrahedrite: Scalable Synthesis and Thermoelectric Performance"

_nanomaterials, 2021, doi:10.3390/nano11061386_

Round 1
Reviewer 1 Report
A very well structured and written piece of work that is of great interest to researchers in the area of nanostructured materials.
Prior to re-submission, a minor spell check is required and grammar check by a qualified English speaker.
Also, Figures 1, 3, 4, 7 and 8 are not very clear. Can you supply high resolution images prior to re-submission?
Author Response
1) A very well structured and written piece of work that is of great interest to researchers in the area of nanostructured materials. Prior to re-submission, a minor spell check is required and grammar check by a qualified English speaker.
Answer: A qualified checking of English language used in the manuscript has been performed by postdoc born in Great Britain. His PhD study was aimed in the topic close to the one treated in the manuscript so he was qualified not only to check English grammar but also the scientific content of the manuscript.
2) Also, Figures 1, 3, 4, 7 and 8 are not very clear. Can you supply high resolution images prior to re-submission?
Answer: All Figures were supplied to the editorial office in the required quality. However, in version supplied to us as well as to reviewers from MDPI, the quality of Figures was worsen. We consulted and cleared the situation with the editorial office. Moreover, we further modified the description of the figures.
Reviewer 2 Report
The authors study the effect of Bi doping on thermoelectric properties of tetrahedrite. They perform structural and thermoelectrical characterization of synthesized materials using SEM, TEM, XRD, XPS, and Seeback coefficient measurement systems. It is shown that increasing Bi content on tetrahedrite Cu12Sb(4-x)BixS13 (x=0.02-0.20) is correlated with lower thermal conductivities.
I would suggest a significant revision before the paper being accepted for the Nanomaterials journal.
- Most of the figures are in low resolution especially Figures 1&2 were not comprehensible.
- English of the text needs to be revised, sentences like "Especially 61 a Portuguese group[33-36] paid a big attention to Bi doping of Cu12Sb4S13." are not acceptable in academic papers.
- Conclusions drawn from TEM/EDS analysis (i.e. Cu enrichment at grain boundaries) are not thoroughly backed up by EDS maps.
- It is not clear the location of the presented point spectra in Fig.5 and Fig.6.
- To provide a Bismuth elemental mapping of either SEM or TEM analysis in the main text or SI would significantly support the claim of authors on higher Bi concentration in skinnerite phase.
- I do not agree with the interpretation of TEM image in Figure 6. Nanopores usually do not present as bright spots. Additionally, authors should include the SAED images of assigned phases in the main text or SI.
Author Response
1) Most of the figures are in low resolution especially Figures 1&2 were not comprehensible.
Answer: All Figures were supplied to Editorial office in the required quality. However, in version supplied to us as well as to reviewers from MDPI the quality of Figures was worsen. We consulted and cleared the situation with Editorial office. Moreover, we further modified the description of Figures.
2) English of the text needs to be revised, sentences like "Especially 61 a Portuguese group[33-36] paid a big attention to Bi doping of Cu12Sb4S13." are not acceptable in academic papers.
Answer: a qualified checking of English language used in the manuscript has been performed by postdoc born in Great Britain. His PhD study was aimed in the topic close to the one treated in the manuscript so he was qualified not only to check English grammar but also the scientific content of the manuscript.
3) Conclusions drawn from TEM/EDS analysis (i.e. Cu enrichment at grain boundaries) are not thoroughly backed up by EDS maps.
Answer: for TEM analyses we used a conventional TEM (JEM 2100) which is not equipped with a scanning unit; therefore EDS mapping is not possible and our measurements were limited to point EDS analysis. We agree with the reviewer that the issue of Cu enrichment at GBs is not investigated thoroughly enough (we measured only few GBs) therefore we decided to omit this statement from the manuscript, also because this issue is also not central to the topic of the manuscript which is related to Bi doping.
4) It is not clear the location of the presented point spectra in Fig.5 and Fig.6.
Answer: we added points of EDS analyses in SEM and TEM. In SEM/EDS, the marked points do not exactly denote the area of analysis due to the interaction volume that extends into the sample depth and is usually in the range of few microns. In this sample, this usually exceeds the average size of particles, therefore quantification of SEM/EDS spectra is only approximately described in this paper. Newertheless, it was possible to distinguish between the three main phases by analyzing largest possible particles with uniform contrast.
We also added areas of TEM/EDS analyses in Figure 6. In TEM, the effect of interaction volume is much less relevant because of the small thickness of the sample (up to 100 nm). In TEM we performed the analyses of single grains using electron beam with diameter of up to 10 nm in order to avoid beam damage due to the interaction between high-energy beam and the sample when using focused beam.
5) To provide a Bismuth elemental mapping of either SEM or TEM analysis in the main text or SI would significantly support the claim of authors on higher Bi concentration in skinnerite phase.
Answer: we have tried to map Bi in SEM but the results were not satisfactory because of the low overall amount of Bi in the sample (nominal composition Cu12Sb3.8Bi0.2S13). Even in skinnerite grains, where Bi is concentrated, its amount is slightly above the EDS detection limit. Also, the size of most grains is too low to obtain reasonable results. Therefore, we decided to limit EDS measurements in SEM to point analyses on larger grains, which gave consistent results. With SEM/EDS Bi was only detected inside skinnerite grains (grains with brightest contrast). In SEM images recorded at higher magnifications it is possible to observe very fine (less than 100 nm) particles with brighter contrast, which could be Bi rich. However, such particles are too small for analyses with SEM. The only useful tool for analysis of such small particles is TEM.
6) I do not agree with the interpretation of TEM image in Figure 6. Nanopores usually do not present as bright spots.
Answer: there figures are conventional bright-filed TEM images - the images include diffracted beams and also the direct beam (these are not Z-contrast STEM images, where the hole and pore would show dark contrast and areas of higher average Z would be brighter). In conventional TEM images, areas without the sample are brightest, because these are formed only by non-scattered electrons (direct beam), see lower left part of the image - hole - in Fig. 6a (the hole is also marked on Figure S2. Pores in such images show similar contrast as the hole (bright) contrast because electrons are not scattered in these areas without material to scatter electrons.
7) Additionally, authors should include the SAED images of assigned phases in the main text or SI.
Answer: we have added additional TEM image of the sample with the corresponding experimental SAED pattern of a tetrahedrite grain oriented along its [111] zone axis in the Supplemental Materials (Figure S2). The majority of grains in the TEM sample belonged to tetrahedrite, only few grains of skinnerite were found, whereas famatinite grains were not found in the TEM sample. In order to record diffraction pattern of a certain grain, it must be oriented close to a low-index zone axis. In polycrystalline samples with random grain orientation it might challenging to find a grain close to some zone axis. In our sample that contained mostly tetrahedrite grains we were able to find a tetrahedrite grain oriented close to the [111] zone axis as shown in Figure S2. However, we did not find any skinnerite grain oriented close to a low-index zone axis, therefore we did not record SEAD patterns form these grains. Identification of tetrahedrite and skinnerite grains in TEM is based mainly on EDS analyses: (i) skinnerite grains have S : Cu ratio of 1, whereas in tetrahedrite, the ratio is >1 and (ii) skinnerite grains contain significant amount of Bi, whereas Bi was not detected in tetrahedrite.
Reviewer 3 Report
Recommendation: minor revision
Comments:
This work systematically studied the effect of Bi doping on the microstructure evolution of tetrahedrite Cu12Sb4S13 and the concomitant thermoelectric properties. It is found that increasing Bi content can promote the formation of skinnerite Cu3SbS3 and famatinite Cu3SbS4, while decreasing the content of matrix tetrahedrite phase. The presence of impurity phases decreases the thermal conductivity but as well the electrical conductivity, and even the latter overwhelms the former. As a result, the final zT is decreased by doping Bi. Even though the thermoelectric performance is not enhanced by the doping strategy, I found the research logic is clear and the results can also provide some useful discussions on the “double-edged sword” effect of chemical doping and phase separation. I will recommend the publication of this work after addressing my following minor comments.
- It is clear that the authors are not the first ones who study Bi doping in this materials systems. Is the conclusion in this work consistent with other reports? If no, why?
- How does the oxidation state of Bi influence the phase formation of secondary phases?
- The thermoelectric properties are dominated by interfaces. The authors claimed slightly increased Cu at some grain boundaries. Will this chemical decoration of GBs impact the transport properties?
- The data points of electrical properties show strange fluctuations at adjacent temperatures in Figure 7. Why?
- What causes the inverse “U” shape of temperature-dependent electrical conductivity? Could it be a GB or interface effect?
- In my opinion, this work is a good example to show that not all complex microstructures are beneficial to thermoelectric properties. A tradeoff between electron and phonon transport must be meticulously designed. The authors could add more discussions on this viewpoint.
- The image qualities are too low which makes details fuzzy.
Author Response
1) It is clear that the authors are not the first ones who study Bi doping in this material systems. Is the conclusion in this work consistent with other reports? If no, why?
Answer: as claimed in Introduction, there is a gap in understanding of Bi presence in Cu12Sb4-xBixS13 system. While in paper [39] interval of x=0.2-0.8 was studied, in paper [38] interval x=0.1-0.4 was applied. In both cases the result was “less Bi, better thermoelectricity“. These experiments opened door to study influence of Bi doping in the range x=0.02-0.20 as manifested in our case. The results fit very well with older findings about influence of Bi doping on deteriation of tetrahedrite thermoelectric properties. However, the main bonus is the elucidation of Bi accumulation presence in skinnerite which itself (without Bi) does not belong to good candidates for thermoelectric materials. Moreover, the difference is also in fact, that doping as the traditional way to enhance ZT values [14] was performed by a scalable procedure using combination of non-equilibrium high-energy milling and SPS treatment. This is an unique feature presented in manuscript in comparison with other papers [33-40] performed in a laboratory scale.
2) How does the oxidation state of Bi influence the phase formation of secondary phases?
Answer: the occurence of the individual phases in Cu-Sb-S system composed of tetrahedrite, famatinite and skinnerite is influenced by Bi addition. Its increasing amount in the range x =0.02-0.20 favors formation of famatinite and skinnerite, while formation of tetrahedrite is decreasing. Formally, antimony for which Bi has to be substituted is in 3+ state in tetrahedrite and skinnerite, while in famatinite is in 5+ state. In most heavily doped sample (x=0.20) the studies performed by XPS method revealed presence of Bi in 3+ state as well as in elemental form. Using TEM/EDS method it was revealed that skinnerite grains contain significant amount of Bi, whereas Bi was not documented in tetrahedrite.
3) The thermoelectric properties are dominated by interfaces. The authors claimed slightly increased Cu at some grain boundaries. Will this chemical decoration of GBs impact the transport properties?
Answer: It is an interesting question. Yes, the presence of Cu at GB can impact the transport pp. However, the presence of different phases with various compositions also affects the transport properties. So, it is difficult to establish some clear conclusions about the role of GB.
4) The data points of electrical properties show strange fluctuations at adjacent temperatures in Figure 7. Why?
Answer: The fluctuations are explained by the fact that some points correspond to heating or cooling cycle. We apologize to have omitted this point. It is now indicated on the figure. The irreversibility is due to a slight reaction between phases during measurements.
5) What causes the inverse “U” shape of temperature-dependent electrical conductivity? Could it be a GB or interface effect?
Answer: The U shape is classically observed in tetrahedrite presenting such Seebeck coefficient. The material is in intermediate regime between degenerate SC and SC. This point is added in the discussion.
6) In my opinion, this work is a good example to show that not all complex microstructures are beneficial to thermoelectric properties. A tradeoff between electron and phonon transport must be meticulously designed. The authors could add more discussions on this viewpoint.
Answer: We do agree with the reviewer. This sentence has been added in the conclusion. “This illustrates the degradation effect of Bi doping on the thermoelectric properties of tetrahedrite as well as demonstrating that complex microstructures are not always beneficial for the thermoelectric performance. A trade-off between electrical conductivity and Seebeck coefficient must be considered in order to retain a competitive power factor and thus benefit from a lower thermal conductivity. In the present case, significant amounts of second phases are very detrimental to the thermoelectric performances despite interesting microstructural features.”
7) The image qualities are too low which makes details fuzzy.
Answer: all Figures were supplied to the editorial office in the required quality. However, in version supplied to us as well as to reviewers from MDPI the quality of Figures was worsen. We consulted and cleared the situation with the editorial office. Moreover, we further modified the description of Figures.
Round 2
Reviewer 2 Report
I would like to thank the authors for the edits, explanation provided in their response and additional data provided in SI. I am happy to suggest acceptance of the manuscript without additional revisions.